# OpenReview forum: "QuadSentinel: Sequent Safety for Machine-Checkable Control in Multi-agent Systems"
_ICLR.cc/2026/Conference — Submitted to ICLR 2026_

### Official Review · Reviewer_xhBP · 2025-11-01

**Soundness:** 2
**Presentation:** 2
**Contribution:** 3
**Rating:** 4
**Confidence:** 4

**Summary:**

The paper proposes QuadSentinel, a modular multi-agent guard framework designed to provide machine-checkable safety control for autonomous multi-agent systems. Specifically, the method translates safety policies into formal logical rules expressed as sequents over boolean predicates grounded in observable state. The system consists of four cooperating agents, i.e., a state tracker, policy verifier, threat watcher, and hierarchical referee, which jointly monitor inter-agent messages and actions in real time, selectively evaluate relevant predicates via a top-k retrieval mechanism to reduce cost, and issue allow/deny decisions with auditable justifications. Experiments on ST-WebAgentBench and AgentHarm show that QuadSentinel achieves higher accuracy, precision, and recall with lower false-positive rates than prior guardrails such as ShieldAgent and GuardAgent, and ablations demonstrate that each component (hierarchical referee, threat watcher, predicate filter) contributes meaningfully to overall performance.

**Strengths:**

1. The paper proposes a novel multi-agent guard framework that translates natural-language safety policies into propositional logics and enforces them through coordinated agents, extending prior single-agent guardrails e.g. ShieldAgent, and offering a clear conceptual contribution to runtime safety in multi-agent systems.

2. The proposed method achieves good performance on the evaluated benchmarks, where it consistently outperforms baselines on multiple safety benchmarks, and the ablation study shows the individual contributions from each system component.

**Weaknesses:**

1. The framework relies heavily on the correctness of the offline policy-to-rule translation step, but the paper provides limited quantitative evaluation of translation fidelity or failure cases, leaving uncertainty about robustness when policies are ambiguous or domain-shifted.

2. Although the method is positioned as low-overhead, the reported runtime analysis in the appendix is theoretical, and no actual latency, throughput, or cost measurements are provided for real deployments, especially under high-frequency multi-agent interaction loads.

3. The overall pipeline is highly complex, with multiple sub-agents where each has individual components and heuristics, however the key design details are insufficiently explained in both the main text and appendix. For example, how is the “must-check” set chosen in the state tracker? How do you determine the “high-precision evaluator,” and is there standalone validation of its correctness? The authors claim they only need to update only a subset of predicates each step yet provide no justification for robustness under partial predicate observation or error accumulation. It would be interesting to see an analysis on the guard decision accuracy vs cost while varying the evaluation budget.

4. The presentation lacks clarity due to frequent mixing of language descriptions and mathematical notation, which disrupts flow and makes it harder for readers to follow the core methodology.

**Questions:**

1. How sensitive is QuadSentinel to errors in the offline policy transition stage? Can the authors provide more case studies of the policies and extracted rules, and also provide quantitative results to justify that the system remains reliable when the NLP policies are noisy, incomplete, or adversarially phrased?

2. The evaluation focuses on safety effectiveness but not runtime feasibility. Can the authors report concrete latency and token-cost measurements (per decision, per agent step) under realistic multi-agent workloads, and compare them directly with single-guard baselines to validate the “low-overhead” claim beyond theoretical complexity analysis?

---

> ### Author Response · Authors · 2025-11-23
> **Response to Reviewer xhBP**
>
> We sincerely thank the reviewer for the time and effort spent evaluating QuadSentinel. We value the assessment that our work is a "novel multi-agent guard framework" that makes a clear conceptual contribution. We also appreciate the constructive criticism regarding presentation clarity and the need for concrete runtime analysis.
>
> We have taken these concerns to heart and have conducted additional experiments during the rebuttal period to address the reviewer's questions regarding latency and robustness. We provide detailed responses below.
>
> ### **1\. Runtime Feasibility: Concrete Latency and Cost Analysis**
>
> The reviewer correctly pointed out that our claim of "low overhead" lacked empirical backing in the main text. To address this, we profiled the computational cost of QuadSentinel compared to the base Agent and two baselines (ShieldAgent and GuardAgent) using GPT-4o-mini on the AgentHarm benchmark.
>
> | System Component | Additional Token Cost (Overhead) | Additional Time Cost (Overhead) |
> | ----- | ----- | ----- |
> | *Base Agent (Reference)* | *(3.9M)* | *(10,699s)* |
> | **QuadSentinel (Ours)** | **3.7M (\~0.95x)** | **3,550s (\~0.33x)** |
> | ShieldAgent | 7.3M (\~1.86x) | 6,640s (\~0.62x) |
> | GuardAgent | 10.4M (\~2.66x) | 13,302s (\~1.24x) |
>
> **Findings:**
>
> * **Latency:** QuadSentinel introduces the lowest latency overhead (\~0.33x), significantly outperforming ShieldAgent (\~0.62x) and GuardAgent (\~1.24x). This validates that our design is suitable for high-frequency multi-agent interaction loads.
> * **Efficiency:** The system is highly token-efficient due to our "High-Read, Low-Write" architecture. While the guard monitors extensive context, the specialized components (State Tracker, Referee) output very concise decisions, avoiding the expensive long-form generation costs seen in other baselines.
>
> ### **2\. Robustness of Policy Translation**
>
> The reviewer asked about the sensitivity of the system to errors in the offline policy translation stage and how we handle ambiguous or adversarially phrased policies.
>
> * **Offline Verification:** It is important to clarify that policy translation is a one-time, offline process, not a runtime dependency. Because this happens offline, we are not constrained by latency. We can utilize the most powerful reasoning models available (e.g., GPT-5 pro or Gemini-3 pro) to ensure high fidelity in decomposing policies into predicates. Furthermore, this design explicitly supports a human-in-the-loop workflow: experts can review, refine, or correct the compiled logic rules before they are ever loaded into the runtime system.
> * **Design Philosophy:** Our design philosophy is to anchor the system in these verified policies rather than asking an LLM to interpret raw, ambiguous guidelines on the fly. By grounding safety in compiled logic, we minimize the risk of the guard "misinterpreting" the core rules. However, we do not view the system as a rigid automaton. The LLM-based Referee retains the necessary flexibility to interpret complex contexts and handle edge cases that strict boolean logic might miss.
>
> ### **3\. Pipeline Details and Top-k Sensitivity**
>
> The reviewer raised excellent questions regarding the specific mechanics of the "must-check" set and the robustness of partial predicate updates.
>
> * **Must-Check Set:** This set is determined deterministically during the offline compilation. We build a dependency graph linking API tools to specific rules. If an agent invokes a specific tool type (e.g., `send_email`), the antecedent predicates for all rules governing emails are automatically added to the "must-check" list. This ensures that safety-critical predicates are never skipped, regardless of the embedding retrieval score.
> * **Top-k Sensitivity Analysis:** The reviewer asked to see an analysis of decision accuracy vs. evaluation budget. We performed a sensitivity analysis varying k (as a % of total predicates) to find the optimal balance.
>
> | k (% of predicates) | Accuracy | Precision | Recall | FPR |
> | ----- | ----- | ----- | ----- | ----- |
> | k=5% | 89.8% | 94.9% | 84.1% | 4.5% |
> | **k=10% (Optimal)** | **91.5%** | **97.4%** | 85.2% | **2.3%** |
> | k=15% | 89.5% | 93.7% | 84.7% | 5.7% |
> | k=40% | 88.6% | 90.5% | **86.4%** | 9.1% |
> | k=100% (Full) | 88.9% | 92.0% | 85.2% | 7.4% |
>
> **Finding:** The results show that the Top-k mechanism acts as a vital signal-to-noise filter, not just a speed optimization. Providing the verifier with all predicates (k=100%) actually degrades performance (Accuracy drops from 91.5% to 88.9%) by introducing irrelevant context that confuses the model. The "sweet spot" at \~10% provides the clearest signal for accurate verification.

---

> ### Author Response · Authors · 2025-11-23
> **Response to Reviewer xhBP (Cont'd)**
>
> ### **4\. Presentation and Clarity**
>
> We completely agree with the reviewer's assessment that the mixing of formal notation and text in Section 3 disrupted the flow.
>
> We will upload a revision shortly, which includes:
>
> * **Simplified Formalism:** We are rewriting Section 3 to simplify the mathematical notation, keeping only what is necessary to define the sequents.
> * **Concrete Examples:** We are moving the running examples and case studies from the Appendix directly into the main text. This will provide an immediate, intuitive grounding for the State Tracker, Threat Watcher, and Referee components, making the pipeline much easier to follow.
> * **Expanded Details:** We will incorporate the definitions of the "high-precision evaluator" (the LLM call that evaluates the filtered predicates) and the "must-check" logic directly into the methodology section to avoid ambiguity.
>
> We hope these clarifications and new data points address the reviewer's concerns, and we thank the reviewer again for the time and effort dedicated to reviewing our work.

---

> ### Comment · Reviewer_xhBP · 2025-11-27
>
> Thanks to the authors for their detailed response. Most of my concerns have been addressed. However, I still have one minor question regarding how the dependency graph is built, and whether using this graph can guarantee retrieval of all relevant rules for verification of the current action. For instance, there are many indirect rules that do not directly involve the tool-call predicates but still logically influence verification. For example, would the dependency graph for `send_email` include `content_is_harmful`, even though it does not contain the keyword `email`?
>
> Regardless, I believe this is a novel and meaningful contribution to the guardrailing of multi-agent systems, and I have decided to increase my score.

---

> ### Author Response · Authors · 2025-12-02
>
> We thank the reviewer for the continued engagement and for raising the score. We are glad that our additional experiments and clarifications have addressed the primary concerns regarding runtime feasibility and system robustness.
> Regarding the question on the dependency graph and indirect rules (e.g., `send_email` triggering `content_is_harmful`):
> **The dependency graph is based on logical structure and explicit schema registration, not keyword matching.**
> The construction happens offline during the policy translation phase. To use the specific example: if a policy is translated into the rule: `NOT (tool_is_email AND content_is_harmful)`, the dependency is built in three deterministic steps:
>
> 1. **Explicit Schema Binding (Tool $\to$ Rules):** The system builds a hard mapping between API definitions and the rules that govern them. In this case, the compiler identifies that this rule restricts the send\_email API.
> 2. **Logical Extraction (Rules $\to$ Predicates):** It then identifies every predicate required to evaluate this rule. In this case, that includes both `tool_is_email` and `content_is_harmful`.
> 3. **Dependency Building (Tool $\to$ Predicates):** It creates dependency edges from the `send_email` tool to the full set of predicates.
>
> **Conclusion**: Yes, the graph guarantees the retrieval of `content_is_harmful` even though it lacks the keyword “email”. When the tool is invoked, the system follows these edges to populate the Must-Check Set, ensuring `content_is_harmful` is evaluated because it is a logical antecedent of the rule governing the `send_email` tool.
> We appreciate this insightful question, as it highlights the advantage of compiling natural language into structured logic.

---

### Official Review · Reviewer_J6ph · 2025-11-01

**Soundness:** 3
**Presentation:** 3
**Contribution:** 3
**Rating:** 6
**Confidence:** 3

**Summary:**

This paper studies safety issues in multi-agent LLM systems, focusing on real-time protection during agent interactions at the action level. It proposes a structured supervisory framework, QuadSentinel, which compiles deployer-written policies into machine-checkable rules through an offline policy registration and translation stage. At runtime, QuadSentinel enforces safety constraints using a multi-agent system with four guards: state tracker, policy verifier, threat watcher, and hierarchical referee, which provides dynamic action- and trajectory-level checks for other autonomous multi-agent LLM systems. The proposed framework is evaluated on two standard safety benchmarks, ST-
WebAgentBench and AgentHarm, and the experimental results show that QuadSentinel is able to achieve balance performance w.r.t. accuracy, precision, recall, and false positive rate.

**Strengths:**

- This paper addresses a challenging and important problem: safety in multi-agent LLM systems.

- This paper proposes a relatively complete and thorough framework, capable of dynamic online monitoring with action- and trajectory-level safety enforcement for multi-agent systems. It also considers efficiency perspective, e.g., incremental updating of predicates to avoid full re-evaluation, and risk-cost optimization.

- The translation from natural language policies to predicates and action and message rules is inspiring.

- The experimental results demonstrate high accuracy/precision/recall and low false positive rates, which supports the effectiveness of QuadSentinel. Additionally, in the ablation study, it is interesting to observe the significantly higher false positive rate when using a single referee, which may provide insights for future research.

- The paper is well organized, clear, and presented effectively.

**Weaknesses:**

- While QuadSentinel translates natural-language policies into sequents, it is unclear how robust the system is to ambiguous or conflicting policies. It is better to include such clarification or discussion in the paper.

- The framework may fail for risky actions or malicious attacks that are unseen from the registered policy book. It would be beneficial for the paper to discuss potential adaptivity or mitigation strategies for unseen threats.

- It would be better to show some case studies.

- It would be better to show the latency analysis.

**Questions:**

- Since safety checks and the referee operate at the action level, how might blocking certain actions affect agent interactions and overall task performance?

- For top-k filtering, how is k chosen? Is there any study or hyperparameter analysis regarding k?

---

> ### Author Response · Authors · 2025-11-23
> **Response to Reviewer J6ph**
>
> We sincerely thank the reviewer for the thoughtful and constructive feedback. We are grateful for the positive assessment of QuadSentinel, particularly for recognizing the framework as "relatively complete and thorough". We deeply appreciate the acknowledgment of the importance of addressing safety in multi-agent systems and the reviewer's observation that our experimental results effectively demonstrate the balance between accuracy and false positive rates.
>
> We have addressed the reviewer's specific questions regarding latency, hyperparameters, and robustness below. All new experiments and data points presented here were collected from the AgentHarm benchmark using GPT-4o-mini.
>
> ### **1\. Latency Analysis and Computational Cost**
>
> The reviewer suggested including a latency analysis to validate the efficiency of the framework. We have conducted a detailed profiling of QuadSentinel compared to the base Multi-Agent System and other baselines.
>
> | System Component | Additional Token Cost (Overhead) | Additional Time Cost (Overhead) |
> | ----- | ----- | ----- |
> | *Base Agent (Reference)* | *(3.9M)* | *(10,699s)* |
> | **QuadSentinel (Ours)** | **3.7M (\~0.95x)** | **3,550s (\~0.33x)** |
> | ShieldAgent | 7.3M (\~1.86x) | 6,640s (\~0.62x) |
> | GuardAgent | 10.4M (\~2.66x) | 13,302s (\~1.24x) |
>
> **Conclusion:** QuadSentinel adds approximately 33% time overhead, which is highly efficient compared to methods that require full serial verification or code generation (like GuardAgent, which adds \~124%). Furthermore, while our system reads significant context, it generates very few output tokens (decisions/scores), resulting in high token efficiency relative to the base agents.
>
> ### **2\. Selection of k (Top-k Filtering)**
>
> Regarding the reviewer's query on the selection of k for top-k filtering, we performed a comprehensive sensitivity analysis varying k as a percentage of total predicates to understand the impact on performance.
>
> | k (% of predicates) | Accuracy | Precision | Recall | FPR |
> | ----- | ----- | ----- | ----- | ----- |
> | k=5% | 89.8% | 94.9% | 84.1% | 4.5% |
> | **k=10% (Optimal)** | **91.5%** | **97.4%** | 85.2% | **2.3%** |
> | k=15% | 89.5% | 93.7% | 84.7% | 5.7% |
> | k=40% | 88.6% | 90.5% | **86.4%** | 9.1% |
> | k=100% (Full) | 88.9% | 92.0% | 85.2% | 7.4% |
>
> **Finding:** The data reveals a "sweet spot" around 10%.
>
> * **Low k (5%):** The system misses relevant context, lowering recall.
> * **High k (40-100%):** Providing excessive irrelevant information introduces "noise," which confuses the verifier and degrades Precision (increasing False Positives). This confirms that the retrieval logic serves as a vital signal-to-noise filter, not just a performance optimization.
>
> ### **3\. Case Studies**
>
> To address the reviewer's suggestion for more concrete examples, we will move several case studies from the appendix to the main text to better illustrate the pipeline. We will also add more case studies to the appendix. We will update the paper shortly to reflect these changes.
>
> ### **4\. Impact of Blocking on Task Performance**
>
> Regarding the reviewer's question on how blocking actions affects agent interactions and task performance, we clarify that in our empirical study, we employed a strict success measure: if a guard blocks a necessary benign action, it counts as a task failure. Our high Benign Accuracy confirms that valid tasks are rarely disrupted under this strict metric.
>
> However, beyond the benchmark metrics, QuadSentinel is designed to minimize task disruption through feedback loops:
>
> * **Rationale as Feedback**: When the Referee blocks an action, it does not simply halt execution. It provides a natural language rationale (e.g., *"Cannot publish API keys externally"*).
> * **Agent Recovery**: This rationale is returned to the acting agent as an observation. This allows the agent to update its plan and attempt a compliant alternative (e.g., removing the key before sending the file), rather than entering a failure state. This feedback mechanism preserves the collaborative flow of the Multi-Agent System while enforcing safety.

---

> ### Author Response · Authors · 2025-11-23
> **Response to Reviewer J6ph (Cont'd)**
>
> ### **5\. Robustness to Ambiguity and Unseen Threats**
>
> Finally, we address the reviewer's concern regarding the robustness of the system to ambiguous policies and unseen threats:
>
> * **Ambiguity:** We handle ambiguity through a "human-in-the-loop" compatible design. The offline translation separates policy interpretation from enforcement. This allows experts to review compiled rules before deployment. Furthermore, the LLM Referee acts as a semantic safety net during runtime, resolving residual ambiguities that the strict logic checks might miss by using the Threat Watcher context as a tie-breaker.
> * **Unseen Threats:** To mitigate threats not covered by the registered policy book (unseen attacks), the Threat Watcher module tracks behavioral anomalies (e.g., aggressive probing or repetitive syntax) independent of specific rule violations. This allows QuadSentinel to flag and throttle high-risk agents based on *behavioral patterns* even if no specific compiled rule is triggered.
>
> We hope these revisions and responses adequately address the reviewer's concerns. We thank the reviewer again for the time and effort dedicated to reviewing our work.

---

### Official Review · Reviewer_X7Kp · 2025-11-01

**Soundness:** 2
**Presentation:** 2
**Contribution:** 1
**Rating:** 2
**Confidence:** 3

**Summary:**

The paper develops a multi-agent guardrail framework to safeguard agent framework. The framework consists of state tracker, threat watcher, policy verifier, and referee.

**Strengths:**

The guardrail of agent system is an interesting research direction. The paper develops a SOTA result solution.

**Weaknesses:**

1. Novelty is limited. The paper also does not clarify the fundamental difference to ShieldAgent well. ShieldAgent uses probablistic inference to compute sort of "rule violation" score, while this paper uses LLM in the loop to track related rules, observe violations and make guardrail predictions. If I understand correctly, this is a fuzzy version of ShieldAgent with LLM in the loop.

The state tracker part may make it more efficient to only consider related rules with LLM as filter, but the efficiency needs to be evaluated empirically as LLM inference is non-trivial. Therefore, there is a cost by LLM and cost by rule traverse tradeoff that should be explrored more.

2. The presentation is not good. The paper seems make the method description too complex with a lot of natations. I do not see the reason why it is made too formal, considering the principle of method is not that mathematic. At least a short paragraph of high-level overview with a running example to show functions of four components can be helpful.

3. Calling the defense a multi-agent framework is questionable. The process seems simple with sequential LLM tool callings. There is no complex LLM workflow or tool usages or much communication across agents. It would be more appropriate to call it LLM-based guardrail (with multiple LLM callings).

4. Experiment results on more advanced real-world agents such as ChatGPT-Agent, Codex, Claude Code would be more interesting than AWM agents.

**Questions:**

Does the evaluation use the same set of policies compared to ShieldAgent?

---

> ### Author Response · Authors · 2025-11-23
> **Response to Reviewer X7Kp**
>
> We thank the reviewer for acknowledging that safeguarding agent systems is an important research direction. However, we respectfully disagree with the reviewer about the novelty of our paper and point out that there is a misunderstanding about the definition of a Multi-Agent System.
>
> ### **1\. Clarification on Novelty and Distinctions from ShieldAgent**
>
> We respectfully disagree with the statement that ‘QuadSentinel is a fuzzy version of ShieldAgent’. We point out three fundamental differences between our method and ShieldAgent:
>
> * **Single-Agent vs. Multi-Agent:** ShieldAgent is a guardrail designed for a single agent triggered *only* by tool invocations. Consequently, it is structurally blind to the communication layer that defines Multi-Agent Systems (MAS). It cannot detect risks in the coordination phase (e.g., prompt injections, collusion, or malicious instructions passed between agents) unless a tool is called. QuadSentinel is explicitly designed for MAS. It intercepts and verifies inter-agent messages. Because message exchanges occur much more frequently than tool calls, we optimize for this dense verification load using a State Tracker with incremental updates, ensuring that per-step checks remain low-overhead. This allows us to stop attack chains at the source *before* they manifest as unsafe actions, which is a capability ShieldAgent cannot provide.
> * **Symbolic Representation vs. Probabilistic Inference:** ShieldAgent relies on Probabilistic Rule Circuits and Markov Logic Networks to calculate a *likelihood* of safety. It is inherently "fuzzy" because the policies themselves are treated as weighted probabilistic nodes. In contrast, QuadSentinel represents policies as formal logical sequents over discrete boolean predicates. While we utilize an LLM as a referee, its decision is structurally constrained to verify a specific witness (the set of violated predicates). This ensures decisions are grounded in explicit evidence rather than a latent probability score, enabling the auditability required for safety-critical systems.
> * **State Management (Dynamic Top-k vs. Static History):** ShieldAgent uses a memory module that retrieves past workflows and re-evaluates the full context for every action. QuadSentinel introduces a State Tracker that maintains a persistent "World State." Crucially, we introduce the Top-k Predicate Updater, which updates only the most relevant predicates based on embeddings while carrying forward the rest under a closed-world assumption. This novel mechanism allows our framework to scale efficiently to large policy sets without processing the full context at every step, which is a capability ShieldAgent lacks.
>
> In summary, characterizing QuadSentinel as a 'fuzzy version' of ShieldAgent fundamentally misinterprets our contribution. Where ShieldAgent relies on probabilistic estimation within a single-agent scope, QuadSentinel enforces precise, witness-based logic across a decentralized multi-agent architecture. These are not incremental adjustments, but necessary architectural shifts that enable the reliable control of complex agent teams in ways ShieldAgent’s design inherently precludes.
>
> ### **2\. The Definition of "Multi-Agent System"**
>
> The reviewer asserts that our system is merely "sequential LLM tool calling." We respectfully disagree. This conflates *workflow* with *architecture*.
>
> In established literature (e.g., AutoGen), a Multi-Agent System is defined by distinct agents with specialized roles maintaining independent, persistent states while collaborating toward a goal. QuadSentinel fits this definition strictly:
>
> * The Threat Watcher maintains an evolving risk profile state that persists across the trajectory.
> * The State Tracker maintains a persistent "World State" of global predicates.
> * The Referee executes decision logic based on inputs from these distinct agents.
>
> This separation of state and role is fundamentally different from a simple sequential chain-of-thought or function calling where no independent state persists between steps.

---

> ### Author Response · Authors · 2025-11-23
> **Response to Reviewer X7Kp (Cont'd)**
>
> ### **3\. Empirical Efficiency Analysis & The Rule Traverse Trade-off**
>
> The reviewer correctly noted that the efficiency of the State Tracker and the "cost by rule traverse tradeoff" needed empirical evaluation. In response, we conducted new experiments profiling the latency and token cost of QuadSentinel against ShieldAgent, GuardAgent, and a "Rule Traverse" baseline on the AgentHarm benchmark using GPT-4o-mini.
>
> | System Component | Additional Token Cost (Overhead) | Additional Time Cost (Overhead) |
> | :---- | :---- | :---- |
> | *Base Agent (Reference)* | *(3.9M)* | *(10,699s)* |
> | **QuadSentinel (Ours)** | **3.7M (\~0.95x)** | **3,550s (\~0.33x)** |
> | ShieldAgent | 7.3M (\~1.86x) | 6,640s (\~0.62x) |
> | GuardAgent | 10.4M (\~2.66x) | 13,302s (\~1.24x) |
> | Rule Traverse | 20.4M (\~5.23x) | 15,736s (\~1.47x) |
>
> The results decisively resolve the reviewer's concern regarding the tradeoff. A full "Rule Traverse" approach is prohibitively expensive, incurring a massive \~5.23x token overhead. By employing our specialized State Tracker with top-k filtering, QuadSentinel reduces this overhead by over 80% (to \~0.95x), making it significantly more efficient than both the brute-force approach and the ShieldAgent baseline. This confirms that the State Tracker is not just a valid architectural choice but an essential component for feasibility.
>
> ### **4\. Base Agents and Policy Sets**
>
> Regarding the suggestion to use agents like ChatGPT-Agent or Claude Code: This is technically infeasible for inter-agent safety research. Proprietary, closed-source agent ecosystems do not expose the internal message bus (communication between sub-agents). Without access to these messages, no external guard can perform the message-level interception that is central to our contribution. We utilized state-of-the-art open-weights systems (Magentic-One, AWM) specifically because they allow the full observability required for rigorous safety auditing.
>
> Finally, regarding the policy sets: ShieldAgent is not open-sourced, so we could not use their exact policy files. However, we strictly followed the methodology described in their paper to generate the corresponding rules, ensuring the semantic content of the policies remained consistent for a fair comparison.
>
> ### **5\. Presentation Improvements**
>
> We appreciate the feedback on notation. In the revision, which will be uploaded shortly, we will simplify the formalism in Section 3 and move several concrete running examples from the Appendix into the main text to provide an immediate, intuitive understanding of the four components.

---

### Author Response · Authors · 2025-11-25
**Revision Summary**

We thank all reviewers for their constructive feedback. We have uploaded a revised manuscript that incorporates the suggestions to strengthen the empirical validation and clarity of the paper.

**Summary of Key Updates:**

* **Runtime Efficiency Analysis:** We added a concrete empirical profiling of runtime overhead in Table 2 (Section 5.3).
* **Top-k Sensitivity Study:** We included a sensitivity analysis in Table 4 (Section 5.4).
* **Improved Presentation & Clarity:**
  * **Simplified Formalism:** We simplified the mathematical notation in Sections 3 and 4 to be less abstract and more accessible.
  * **Integrated Running Examples:** We moved concrete scenarios (e.g., Data Leakage in Section 3.2) from the Appendix into the main text to intuitively illustrate the functions of the system components.
  * **Additional Case Studies:** We have expanded Appendix A.3 with detailed traces to demonstrate the system's decision-making in complex edge cases.

We believe these revisions directly address the concerns regarding efficiency, robustness, and presentation clarity. And we are happy to address any further questions.

---

### Meta-Review · Area_Chair_yoE7 · 2026-01-05

**Summary:**

While the reviewers see some merits in the proposal of a multi-agent safety guardrail, this submission should not be accepted in its current form due to several major weaknesses, including

1. presentation and clarity
2. reliability/robustness of the proposed method
3. overhead/latency analysis (new results provided during rebuttal)

Overall, this paper requires significant modifications and another round of full review.

**Reviewer Concerns:**

only the latency/overhead part is addressed; the other two major issues are still outstanding

**Reviewer Scores:**

#1 and #3 are common concerns. While #3 could be addressed, #1 requires more discussion and reviewers' endorsement, which is less likely to change during the rebuttal.

---

### Decision · Program_Chairs · 2026-01-26

Reject